# Bioinformatics Analysis of *WRKY* Family Genes in *Erianthus fulvus* Ness

**DOI:** 10.3390/genes13112102

**Published:** 2022-11-12

**Authors:** Haowen Chen, Xuzhen Li, Fusheng Li, Dengyu Li, Yang Dong, Yuanhong Fan

**Affiliations:** 1College of Agronomy and Biotechnology, Yunnan Agricultural University, Kunming 650201, China; 2State Key Laboratory for Conservation and Utilization of Bio-Resources in Yunnan, Yunnan Agricultural University, Kunming 650201, China; 3College of Biological Big Data, Yunnan Agricultural University, Kunming 650201, China; 4Yunnan Plateau Characteristic Agricultural Industry Research Institute, Yunnan Agricultural University, Kunming 650201, China

**Keywords:** *Erianthus fulvus* ness, *WRKY* gene family, structural domains, bioinformatics analysis

## Abstract

One of the most prominent transcription factors in higher plants, the *WRKY* gene family, is crucial for secondary metabolism, phytohormone signaling, plant defense responses, and plant responses to abiotic stresses. It can control the expression of a wide range of target genes by coordinating with other DNA-binding or non-DNA-binding interacting proteins. In this study, we performed a genome-wide analysis of the *EfWRKY* genes and initially identified 89 members of the *EfWRKY* transcription factor family. Using some members of the *OsWRKY* transcription factor family, an evolutionary tree was built using the neighbor-joining (NJ) method to classify the 89 members of the *EfWRKY* transcription factor family into three major taxa and one unclassified group. Molecular weights ranged from 22,614.82 to 303,622.06 Da; hydrophilicity ranged from (−0.983)–(0.159); instability coefficients ranged from 40.97–81.30; lipid coefficients ranged from 38.54–91.89; amino acid numbers ranged from 213–2738 bp; isoelectric points ranged from 4.85–10.06. A signal peptide was present in *EfWRKY41* but not in the other proteins, and *EfWRK85* was subcellularly localized to the cell membrane. Chromosome localization revealed that the *WRKY* gene was present on each chromosome, proving that the conserved pattern WRKYGQK is the family’s central conserved motif. Conserved motif analysis showed that practically all members have this motif. Analysis of the cis-acting elements indicated that, in addition to the fundamental TATA-box, CAAT-box, and light-responsive features (GT1-box), there are response elements implicated in numerous hormones, growth regulation, secondary metabolism, and abiotic stressors. These results inform further studies on the function of *EfWRKY* genes and will lead to the improvement of sugarcane.

## 1. Introduction

Sugarcane is a C4 plant of the grass family [1]. It is used as a raw material to manufacture sucrose and to refine ethanol for biomass energy and livestock feed [2]. Sugarcane has a 12 to 18-month reproductive life, and seasonal and lengthy droughts are unavoidable. Water deficiency directly affects the tillering and elongation of sugarcane at the seedling stage, resulting in lower yields [3]. In China, 80 percent of sugarcane is grown in semi-arid or dry areas where droughts occur frequently and cause significant losses [4,5,6]. As a result, it is critical to thoroughly investigate sugarcane’s drought resistance mechanism, tap its drought resistance genes, and produce exceptional drought-tolerant types. *E. fulvus* Ness is a wild species of Cane grass, a near relative of sugarcane, which grows on high-altitude slopes in arid, barren, and cold environments, even in the cracks of steep rock walls. Therefore, it is exceptionally tolerant of drought, barrenness, and cold compared to other sugarcane varieties [7].

*WRKY* transcription factors constitute one of the most prominent families of transcriptional regulators in plants and are involved in regulating various plant functions through signaling networks [8]. The name of the *WRKY* family comes from the *WRKY* domain, a region consisting of 60 amino acids highly conserved among family members, which is the most noticeable feature of their proteins [9]. The conserved amino acid sequence WRKYGQ(E/K)K and a zinc finger motif at the N terminus of the *WRKY* structural domain serve as its defining features [10]. Since all characterized WRKY proteins bind the identical DNA motif, it is assumed that the WRKY structural domain functions as the DNA binding domain’s single conserved structural component [11]. Every one of the known WRKY proteins has one or two *WRKY* structural domains. The first classification of WRKY proteins was into two major groups [12], with Group I consisting of proteins with two *WRKY* structural domains and Group II consisting of proteins with one *WRKY* structural domain. Members of Groups I and II typically have the same zinc finger structure [13]. Only a tiny part of the WRKY proteins contain zinc finger architectures distinct from Groups I and II. As a result, they are categorized in the newly established Group III. However, it has been demonstrated that individuals from all three groups selectively bind to different W-box elements [14].

Ishiguro and Nakamura identified the first *WRKY* transcription factor, *SPF1*, from sweet potatoes [15]. In recent years, *WRKY* transcription factors have been increasingly analyzed in other plants, such as cucumber [16], tomato [17], and chickpea [18]. As *WRKY* transcription factors have been studied in depth, some of their roles in plants have been gradually explored. *NaWRKY3* in tobacco is required to induce the *NaWRKY6* gene by fatty acid-amino acid conjugates in the oral secretions of the tobacco moth (Manduca sexta) larvae, and silencing one or both of these genes makes plants highly susceptible to herbivore attack. Both genes appear to help plants distinguish between mechanical injury and herbivore attack [19]; Arabidopsis plants with *AtWRKY23* taken out exhibit excellent resistance to encapsulated nematodes because it is up-regulated right away after nematode infection [20]. The *WRKY* gene, isolated from the dry evergreen C3 plant creosote shrub, was first demonstrated to abscisic acid (ABA) signaling in one of the earliest studies [21]. Heat and drought tolerance were improved by heat stress-induced overexpression of *OsWRKY11* mediated by the *HSP101* promoter [21]. And similarly, overexpression of *OsWRKY45* enhanced salt and drought tolerance in addition to disease resistance [22]. Compared to wild-type plants, Arabidopsis plants overexpressing *GmWRKY21* in soybean were more resistant to cold stress, and *GmWRKY54* in soybean was more resistant to salt and drought. Additionally, *WRKY* genes play roles in seed germination and dormancy [23], seed development [24,25], and anti-aging [26,27].

In this study, we comprehensively analyzed the physicochemical properties, secondary structure, conserved motifs, gene structure, cis-acting elements, gene duplication events, and GO enrichment analysis of the *WRKY* gene family in *E. fulvus*. Phylogenetic analysis of the evolutionary relationship of the *WRKY* gene family between *E. fulvus* and rice was performed. Finally, we analyzed the homology of *WRKY* genes in *Arabidopsis thaliana*, *Sorghum bicolor*, *Miscanthus iutarioriparius*, *Oryza sativa*, and *Saccharum spontaneum*. This study will provide valuable information for further studies on the function of *WRKY* genes in E. fulvus and subsequent studies on molecular mechanisms.

## 2. Materials and Methods

### 2.1. Identification of the EfWRKY Gene Family Members

Genomic data, protein files, CDS files, and gff3 files of *E. fulvus* were downloaded from the EfGD: *E. fulvus* Genome Database (https://efgenome.ynau.edu.cn) (accessed on 25 August 2022) online website [28]. The genomes of *Arabidopsis thaliana* (GenBank: GCA_000001735.2), *Sorghum bicolor* (GenBank: GCA_000003195.3), *Miscanthus iutarioriparius* (GenBank: GCA_904845875.1), *Oryza sativa* (GenBank: GCA_001433935.1), and *Saccharum spontaneum* (GenBank: GCA_022457205.1) with gff3 files were downloaded from NCBI. The HMM profile of the *WRKY* conserved structural domain (PF03106) was searched and downloaded from the Pfam database (http://pfam.xfam.org/) (accessed on 3 May 2022). The search command in HMMER 3.0 [29] software and the HMM profile of the *WRKY* conserved structural domain were used to search the sucralose protein files (E-value 10^−5^). A total of 90 *WRKY* genes were searched, and the sequences of the 90 genes in *E. fulvus* were submitted to Batch CD-Search on NCBI (https://www.ncbi.nlm.nih.gov/Structure/bwrpsb/bwrpsb.cgi) (accessed on 26 August 2022) and to the SMART online website (http://smart.embl-heidelberg.de/smart/set_mode) (accessed on 26 August 2022) for structural domain analysis, and genes that did not contain WRKY structural domains were removed, yielding a total of 89 *WRKY* genes. They were renamed *EfWRKY01*–*EfWRKY89*.

### 2.2. Predicted Physicochemical Properties and Secondary Structure Analysis of WRKY Protein in E. fulvus

Prediction of amino acid number (AA), isoelectric point (PI), molecular weight (MW), and hydrophilicity (GRAVY) of WRKY proteins in *E. fulvus* were conducted using the online analysis website (https://web.expasy.org/protparam/) (accessed on 27 August 2022); the Plant-mPLoc server (http://www.csbio.sjtu.edu.cn/bioinf/plant-multi/) (accessed on 28 August 2022) was used for subcellular localization prediction; SignaIP-5.0 (https://services.healthtech.dtu.dk/service.php?SignalP-5.0) (accessed on 28 August 2022) for signal peptide prediction (Parameter setting: organism group is eukaryotic); SOPMA (http://npsa-pbil.ibcp.fr/cgi-bin/npsa_automat.pl?page=npsa_sopma.html) (accessed on 29 August 2022) for analysis of the secondary structure of WRKY proteins in *E. fulvus* (Parameter setting: the number of conception states is 4, and the similarity threshold is 8).

### 2.3. Construction of a Phylogenetic Tree

The WRKY protein sequences of *EfWRKY* genes were subjected to multiple sequence alignment using Jalview software (Mafft with Defaults) and, after construction, were grouped according to the number of WRKY structural domains, the type of zinc finger structures. To clarify the evolutionary relationships between individual members of the *EfWRKY* gene family, 89 *EfWRKYs* genes were compared with 83 *OsWRKYs* genes using the Jalview software (Mafft with Defaults). The results were saved, and a phylogenetic tree was constructed using the neighbor-joining method (NJ) in MEGA 7.0 [30] (Parameter settings: Bootstrap method 1000; p-distance; uniform rates; Pairwise deletion).

### 2.4. Analysis of Conserved Motifs, Gene Structure, and Chromosomal Localization of WRKY Proteins in E. fulvus

Conserved motif prediction for 89 WRKY proteins in *E. fulvus* used MEME [31] (https://meme-suite.org/meme/) (accessed on 27 August 2022); Parameter settings: allowing for recurrence of conserved structural domains and setting the number of conserved motifs to 10; mapping of conserved motifs, gene structure, and gene chromosome positioning of WRKY proteins in *E. fulvus* used TBtools software (for version 0.665).

### 2.5. Chromosome Localization, Gene Duplication, and Covariance Analysis

Two gene amplification mechanisms—tandem duplication and segmental duplication—were considered to examine gene duplication occurrences. The *E. fulvus* genome files were swiftly compared with the gff3 files using the TBtools software (Parameter setting: CPU for BlastP is 2, E-value 10^−5^, Num of BlastHits is 10). Tandem duplications were defined as two gene pairs in the same chromosomal region separated by five or fewer loci. Used TBtools software, Ka (heteronymous substitutions) and Ks (synonymous substitutions) were estimated to evaluate the impact of selection pressure and dispersal time on *EfWRKY* genes (Parameter setting: CPU is 4). To assess the conservation of segmental duplicate gene pairs and direct homologs of *EfWRKY* with other plant species, TBtools software generated visual covariance maps.

### 2.6. Analysis of Cis-Acting Elements of Promoters

A 2000 bp upstream of the gene CDS sequence was extracted and submitted to the PlantCare (http://bioinformatics.psb.ugent.be/webtools/plantcare/html/) (accessed on 27 August 2022)website for cis-acting element prediction, with some unclear elements removed, using TBtools [32] software to draw.

### 2.7. Gene Ontology (GO) Annotation

The Swiss-Prot files were downloaded from the Uniprot website and compared used TBtools software to obtain GO data for 89 *EfWRKY* genes; the GO annotation files for all genes in sucrose were obtained from the EfDG website as background files, and the gene ontology annotations for the *EfWRKY* genes were analyzed using the online website (https://www.omicshare.com/tools/Home/Soft/gogseasenior) (accessed on 10 September 2022).

## 3. Results

### 3.1. Identification and Prediction of Physicochemical Properties of EfWRKY Gene Family Members

89 *WRKY* transcription factors were identified in *E. fulvus* (Appendix A). The 89 WRKY proteins in *E. fulvus* ranged in amino acid length from 213 to 2738 amino acids (aa), molecular weight from 22,614.82 to 303,622.06, the isoelectric point from 4.85 to 10.06, instability coefficient from 40.97 to 81.30, and lipid coefficient from 38.54 to 91.98; all other sub-cells, except for *EfWRKY85*, were localized on the nucleus; all the rest were without signal peptide except *EfWRKY41*, which had a signal peptide (Appendix A).

### 3.2. Secondary Structure Analysis of the89 WRKY Protein in E. fulvus

The study of the secondary structure of the89 WRKY proteins in *E. fulvus* revealed that its secondary structure mainly consisted of irregular coils, α-helices, β-turns, and extended chains, which made it easier to understand the spatial structure of the protein through data analysis. Among these four secondary structure elements, the irregularly coiled coil accounted for the highest percentage, and its function was mainly to connect other secondary structure elements, which ranged from 35.83% (*EfWRKY85*) to 76.67% (*EfWRKY09*). In contrast, the β-turn angle accounted for a smaller percentage of these four structures. The secondary structures of the sucralose *WRKY* family are relatively neat, except for *EfWRKY31*, *EfWRKY61*, and *EfWRKY0*1, which are irregularly curled > extended chain > α-helix > β-turn. In contrast, all other family members are irregularly curled > α-helix > extended chain > β-turn (Appendix A).

### 3.3. Phylogenetic, Conserved Motif, and Gene Structure Analysis of WRKY Gene Family Members in E. fulvus

The 89 WRKY protein sequences of *EfWRKY* genes can be divided into three major groups of five subgroups and one non-class group (Figure 1). The proteins in Group I contain two *WRKY* conserved structural domains (WRKYGQK\WRKYGEK) and zinc finger structures CX4CX22HX1H or CX4CX23HX1H; the proteins in Group II have one *WRKY* conserved structural domain (WRKYGQK/WRKYGKK/WEKFGEK) and a zinc finger structure CX5CX23HX1H. The proteins in Group II were further divided into five subgroups; the proteins in Group III contained one *WRKY* conserved structural domain (WRKYGQK/WRKYGEK) and the zinc finger structure CX5-8CX23HX1C; the remaining three belonged to Group NG. In the evolutionary tree analysis of the *WRKY* and rice *WRKY* gene families in *E. fulvus*, it was found that Group II occupied the majority (50.56%), followed by Group III (30.34%). The minor proportion was Group I (14.61%); the first WRKY structural domain K of the *EfWRKY07* gene was mutated to WRKYGEK, and the same mutations occurred in *EfWRKY31*, *EfWRKY45*, *EfWRKY50*, *EfWRKY54*, and *EfWRKY83*; the conserved WRKY structure of *EfWRKY40*, *EfWRKY53*, *EfWRKY57*, and *EfWRKY67* was mutated to WRKYGKK (Figure 1).This phenomenon is consistent with the variation in *Dendrobium officinale* [33].

Conserved motifs were predicted for 89 WRKY proteins in *E. fulvus* using MEME [31] (https://meme-suite.org/meme/)(accessed on 27 August 2022), allowing for the recurrence of conserved structural domains and setting the number of conserved motifs to 10; distribution of conserved motifs and gene structure analysis revealed that the motif1 motif was highly conserved; two motifs 7 were present in *EfWRKY89*, two motifs 4 in *EfWRKY11*, and only one conserved motif 6 in *EfWRKY87* (Appendix A). The phylogenetic tree among the members of the *EfWRKY* gene family is shown in Figure 2A.There are differences in the conserved motifs between members of the *EfWRKY* gene family, which may also make the functions of the gene family members differ from each other (Figure 2B); Gene structure analysis revealed that all members of the *EfWRKYs* family had introns ranging from 1–9 in number, except for *EfWRKY12*, *EfWRKY14*, and *EfWRKY54*, which had no introns, and some members of the *EfWRKY* gene family did not contain untranslated regions (UTRs) at the 5ʹ end and 3ʹ end (Figure 2C).

### 3.4. Phylogenetic and Functional Analysis of the OsWRKY and EfWRKY Gene Families

To further investigate the different functions of the *EfWRKY* genes, the sequences of the WRKY proteins in *E. fulvus* and *O. sativa* were aligned (Appendix A), and a phylogenetic tree was constructed using the NJ (Neighbor-Joining) (Figure 3). Of the 83 *O. sativa* WRKY proteins, there are 11 with known functions. As can be seen from Figure 3, members of Group I of the *EfWRKY* gene family cluster with *OsWRKY89* due to high protein homology, with the *EfWRKY06* and *OsWRKY89* genes being closely related evolutionarily and likely to play an essential role in biotic and abiotic stress responses [34]; members of *EfWRKY* gene family group IIa clustered with OsWRKY76 and OsWRKY71, and it is possible that members of group IIa in the *EfWRKY* gene family play an active role in plant cold tolerance [35]; members of Group IIb of the *EfWRKY* gene family clustered with *OsWRKY5*, with a strong evolutionary relationship between *EfWRKY62* and the *OsWRKY5* gene, which may have a role in plant leaf senescence [36]; members of Group IIc of the *EfWRKY* gene family cluster with *OsWRKY11*, *OsWRKY29* and *OsWRKY72*, with *EfWRKY13* and *OsWRKY72* genes, being closely related evolutionarily, and may cause endogenous plant jasmonic acid (JA) levels in plants, ultimately causing susceptibility to white leaf blight; there is a close evolutionary relationship between *EfWRKY24* and *OsWRKY29* genes, which may block abscisic acid signaling and inhibit seed dormancy in rice [37]; and there is a close evolutionary relationship between *EfWRKY88* and *OsWRKY11* genes, which may play an essential role in heat and drought stress response and resistance [38]. Members of Group IId of the *EfWRKY* gene family cluster with *OsWRKY13*, with a strong evolutionary relationship between *EfWRKY33* and *OsWRKY13* genes that may have a function in disease resistance in plants [39]; members of Group III of the *EfWRKY* gene family cluster with *OsWRKY47* and *OsWRKY74* are clustered together, with a strong evolutionary relationship between *EfWRKY52* and *OsWRKY47* genes, which may be a positive regulator of water deficit stress response [40], and *EfWRKY18* and *OsWRKY74* genes, which may have a role in plant response to low temperature or in response to iron (Fe) and nitrogen (N) deficiency [40].

### 3.5. Analysis of Cis-Acting Elements of 89 EfWRKY Gene Promoters

In addition to the basic TATA-box, CAAT-box, and light-responsive features (GT1-box), there are also response elements involved in various hormones, growth regulation, secondary metabolism, and abiotic stresses (Appendix A) (Figure 4B).

### 3.6. Chromosomal Localization Analysis of 89 EfWRKY Genes

Chromosomal localization analysis revealed that 89 *EfWRKY* genes were unevenly distributed on ten chromosomes, with the most extensive distribution of 20 *WRKY* genes on Chr4 and the minor distribution of 5 genes on Chr5, 6, and 9; according to Holub [41] et al., chromosomal regions containing two or more genes within 200 kb are defined as tandem duplication events. Tandem and segmental repeats contribute to the expansion of new gene family members and new functions in the evolution of plant genomes. To study gene duplication events in the *E. fulvus* genome, we investigated tandem and segmental duplication during the transition of the *EfWRKY* gene family. A total of 2 *EfWRKY* gene pairs were confirmed to be tandem duplicates. Based on gene duplication analysis, chr1 and chr7 were observed to undergo a tandem duplication gene pair; in addition to the tandem duplication, there were 22 segmental duplications (Figure 5). To determine the selective evolutionary pressure of *EfWRKY* gene divergence after repetition, the Ka and Ks values of the duplicated *EfWRKY* gene pairs were calculated using the KaKs calculator. In general, Ka/Ks = 1 indicates neutral selection, Ka/Ks > 1 indicates positive selection, and Ka/Ks < 1 indicates purifying selection. Ka/Ks < 1 for each duplicated *EfWRKY* gene pair means purifying selection during evolution (Appendix A).

### 3.7. Synteny Relationship of EfWRKY Genes

To further infer evolutionary relationships, *WRKY* genes were compared to identify direct homologous *EfWRKY* gene pairs among *E. fulvus*, *A. thaliana*, *S. bicolor*, *M. iutarioriparius*, *O. sativa*, and *S. officinarum*. Based on the homozygosity analysis, 15 of the 89 *EfWRKY* genes had common lineage pairs in Arabidopsis; 66 had corresponding direct homologs in *O. sativa* and *S. spontaneum*; 70 had related direct homologs in *S. bicolor*, and 71 had corresponding direct homologs in *M. iutarioriparius* (Appendix A). The co-linear plot shows that *EfWRKY* genes are highly evolutionarily homologous to sorghum and manzanita *WRKY* genes, implying that they may have related functions (Figure 6).

### 3.8. Gene Ontology (GO) Annotation

To explore the function of the *EfWRKY* gene in different biological processes, molecular procedures, and cell compartment construction, GO data for 89 *E. fulvus* (Appendix A) were submitted as background files with the EfGD: *E. fulvus* Genome Database (https://efgenome.ynau.edu.cn/index/index/index) (25 August 2022) of all *E. fulvus* gene GO data (Appendix A) as background files submitted online at (https://www.omicshare.com/tools/Home/Soft/gogseasenior) (accessed on 10 September 2022). GO enrichment analysis was performed. As shown in Figure 7, GO function annotation was performed, and, as provided in data s8, gene ontology numbers were also identified during the analysis. Biological processes suggest that *EfWRKY* genes are involved in transcriptional regulation, regulating processes such as biosynthesis and gene expression. Most *EfWRKY* genes are involved in sequence-specific DNA binding functions and transcriptional regulation. It is inferred that they are essential as transcription factors for the growth and development of *E. fulvus* and for regulating the expression of stress-responsive genes.

## 4. Discussion

Crop improvement requires the functional, genomic and structural characterization of individual gene families involved in regulating plant metabolism and developmental processes [42,43]. Although many methods have been developed to describe gene family members, these methods often require considerable effort [44,45,46]. Identifying related gene families and performing careful annotation work is necessary to obtain genes with potentially specialized functions. Many plant genomes have been sequenced, which can be aided by data mining and phylogenetic techniques. In recent years, research has focused on genome-wide analysis of plant gene families, particularly in crops [47,48,49,50], and *WRKY* is one of the larger gene families in plants whose role is regulating plant growth and development [51,52,53]. Similarly, analysis of biotic and abiotic stresses has been demonstrated [54,55,56].

In this study, we identified a total of 89 members of the *EfWRKY* gene family using the pep file of the *E. fulvus* genome and the HMM profile of the conserved structural domain of *WRKY* (PF03106), with individual members having almost identical gene structures (Figure 2), the WRKYGQK heptapeptide structure with the zinc finger structure (C2H2 type or C2HC type). The difference lies in the number of *WRKY* domains. As seen from Figure 2, *EfWRKY37* and *EfWRKY51* have a *WRKY* domain but are assigned to Group I, probably missing during evolution, which usually occurs in monocotyledons [57].

Meanwhile, the WRKYGQK heptapeptide structures of *EfWRKY40*, *EfWRKY53*, *EfWRKY57*, and *EfWRKY67* mutated to WRKYGKK, a variant also found in rice and dates; and the WRKYGQK heptapeptide structure of *EfWRKY31*, *EfWRKY45*, *EfWRKY50*, *EfWRKY54*, and *EfWRKY83* mutated to WRKYGEK. It has been demonstrated that the WRKYGKK structural domain binds specifically to the WK-box, which is significantly different from the WRKYGQK-specific binding [58]. Variants in conserved *WRKY* sequences may affect the regular interaction and binding specificity of *WRKY* genes with downstream target genes, thereby altering gene function [59].

Phylogenetic analysis revealed that the members of the *EfWRKY* gene family conformed to the consistent classification of *WRKY* gene families, except for a few *E. fulvus* WRKY proteins with partially missing or mutated structural domains. All other members had complete and accurate structural parts, indicating to some extent that the *EfWRKY* gene family is relatively evolutionarily conserved. As can be seen from Figure 1, the *EfWRKY* gene family is divided into four groups: Group Ⅰ, Group Ⅱ, Group Ⅲ, and NG; the number of its groups containing *WRKY* genes is 13 (14.61%), 45 (50.56%), 27 (30.34%) and 4 (4.49%), respectively. Group Ⅱ is the group with the highest number of *EfWRKY* genes, indicating that the smallest number of Group IIa *WRKY* genes may be due to its relatively short evolutionary time compared to the other groups, being the last of all groups in the *EfWRKY* gene family to evolve. This explains the variation and deletion of conserved structural domains of WRKY proteins in *E. fulvus*, suggesting that *E. fulvus* may be gaining evolution.

In conclusion, systematic analysis of the *EfWRKY* gene family could provide a valuable reference for more detailed functional genomic characterization of these genes and help to predict and select suitable candidate genes for sugarcane improvement. As more complete sequenced genomes become available, more *WRKY* genes will likely be identified in other plants. Based on these data, a more comprehensive understanding of the origin and evolution of the *WRKY* gene family will probably be found in the data identified, and its functional characteristics can be obtained.

## 5. Conclusions

In conclusion, we performed a genome-wide identification of *WRKY* gene families in cane grass. Using bioinformatics tools, we investigated their evolutionary relationships, gene structure, replication events, and cis-acting promoter progenitors. Eighty-nine *WRKY* genes with *WRKY* structural domains were identified in the *E. fulvus* genome and distributed unevenly on ten chromosomes. Based on sequence alignment and phylogenetic analysis, *EfWRKY* genes were classified into three prominent families, five subgroups, and one non-main family. The gene structure and conserved motif analysis were also consistent with the phylogenetic classification. The gene structure and motif patterns indicated that *EfWRKY* members in the same subgroup showed extensive similarity. Tandem and segmental duplication significantly contributed to the expansion of the *WRKY* gene family in *E. fulvus*. Evolutionary divergence analysis (Ka/Ks) indicated that *EfWRKYs* were subject to strong purifying selection during plant evolution. Homology analysis showed that 70 and 71 *EfWRKY* genes were directly homologous to *S. bicolor* and *M. iutarioriparius*, respectively. In addition, subcellular localization revealed that 88 *EfWRKYs* genes were mainly located in the nucleus. Overall, our findings describe the evolutionary characteristics, genomic replication events, and molecular biological functions of *WRKY* genes in cane grass. In conclusion, our results will provide a basis for elucidating the molecular mechanisms and further functional characterization of the *EfWRKY* gene family, thus providing a resource for plant breeding and genetic engineering.

## Figures and Tables

**Figure 1 genes-13-02102-f001:**
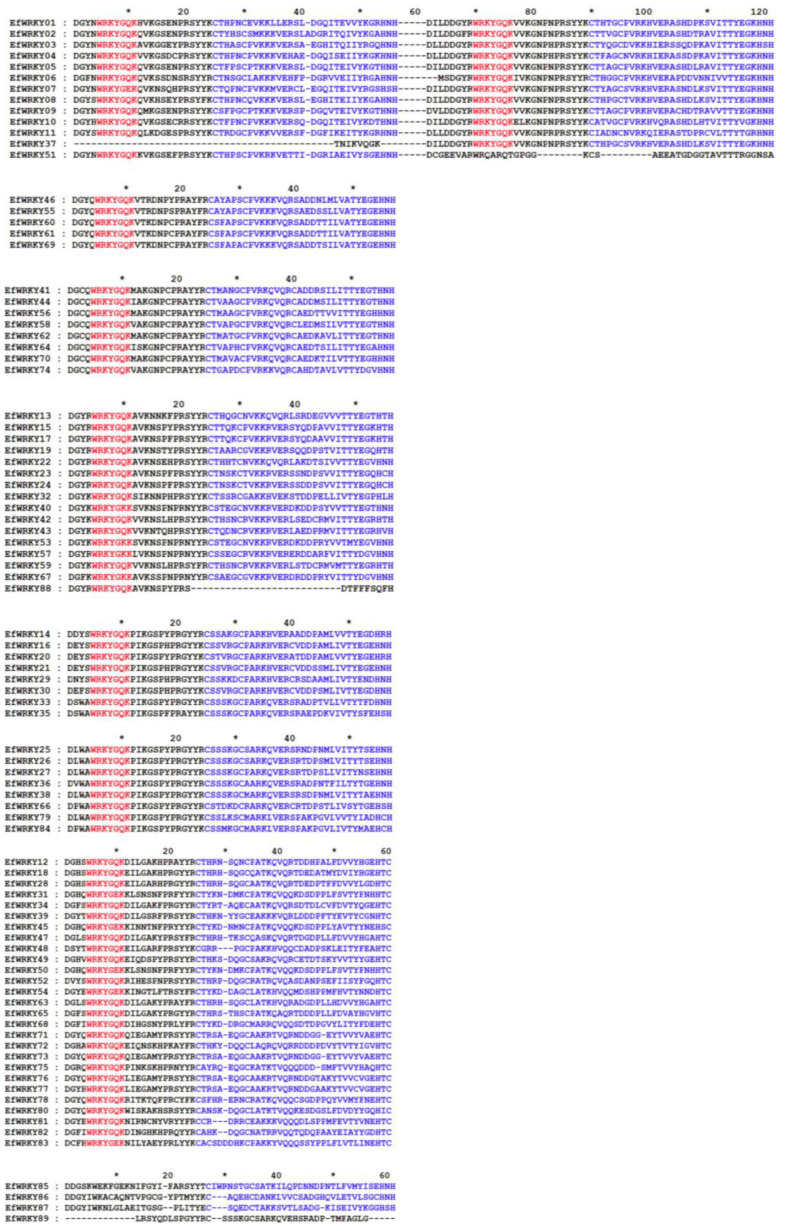
Sequence alignment of the WRKY structural domain of the 89 WRKY proteins in *E. fulvus*. The red part is the highly conserved heptapeptide of the protein, and the blue part is the zinc finger structure of the protein.

**Figure 2 genes-13-02102-f002:**
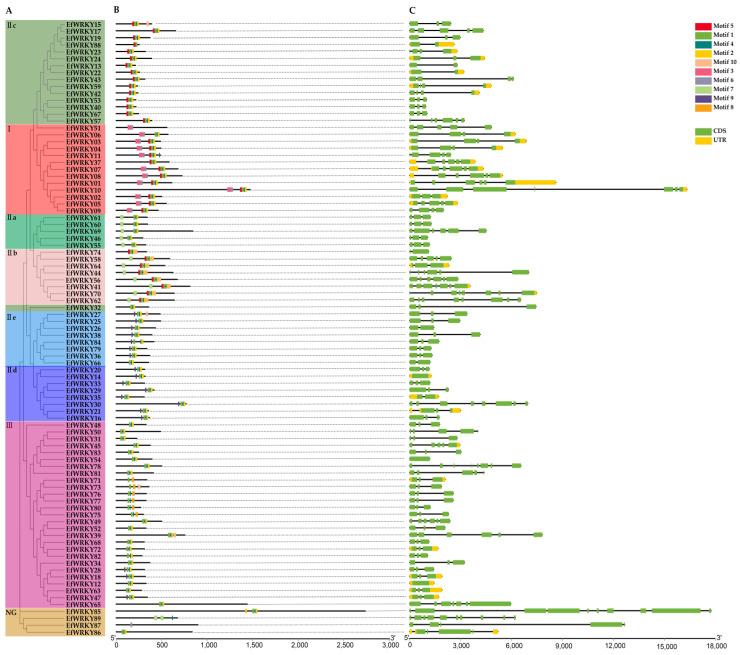
Conserved motifs and gene structure maps of 89 *EfWRKY* genes. (**A**) Phylogenetic tree of 89 *EfWRKY* genes. (**B**) Motif composition of 89 *EfWRKY* genes; a total of 10 types of conserved domains were found. The black line indicates the protein length. (**C**) Gene structure of 89 *EfWRKY* genes. Black lines indicate introns, green boxes indicate CDS, and yellow boxes indicate UTR.

**Figure 3 genes-13-02102-f003:**
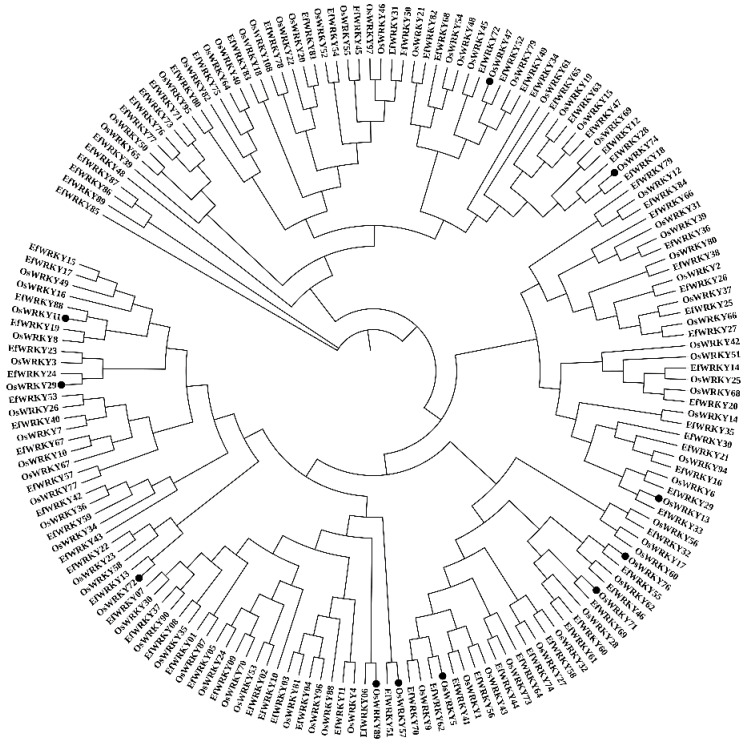
Phylogenetic tree of WRKY protein sequences in *E. fulvus* and some *O. sativa*. (● indicates known *WRKY* genes in rice).

**Figure 4 genes-13-02102-f004:**
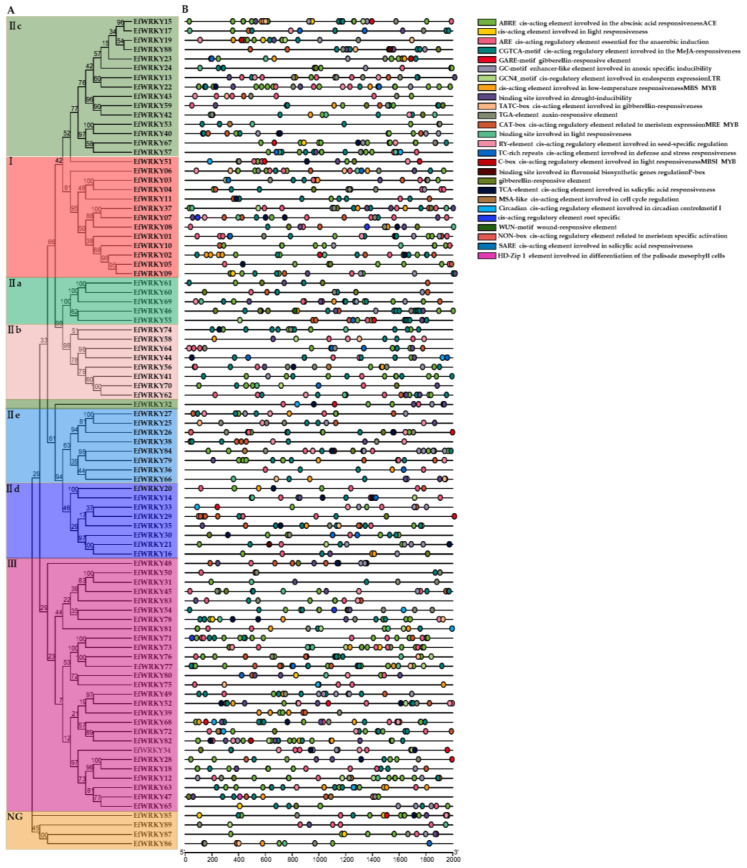
Cis-element analysis of the *EfWRKY* gene family. Different colors represent a total of 26 types of functional modules. (**A**) Phylogenetic tree of 89 *EfWRKY* genes. Numbers on branches represent bootstrap estimates for 1000 replicate analysis or clade credibility values. (**B**) Cis-acting elements.

**Figure 5 genes-13-02102-f005:**
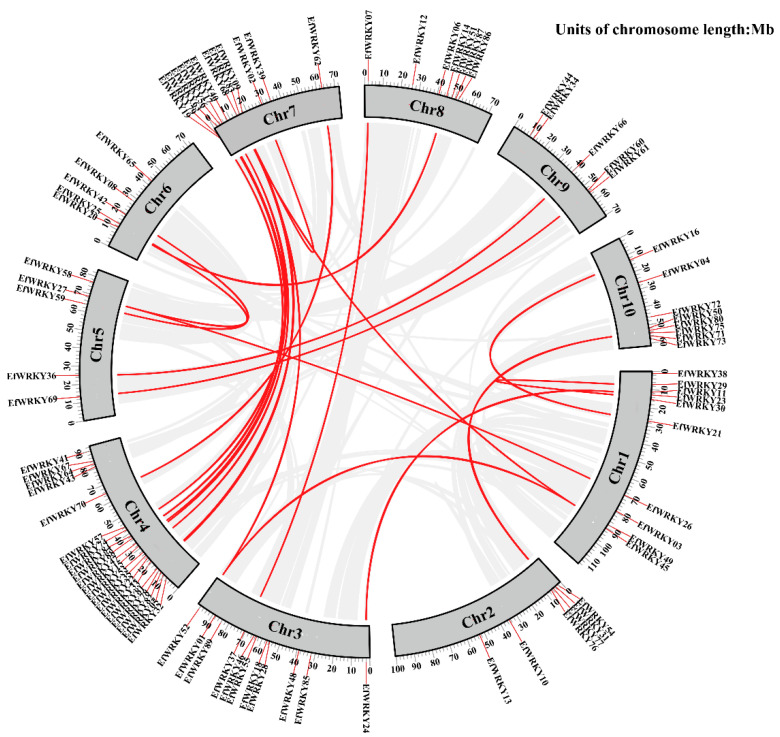
The intrachromosomal segmental duplication map of the *WRKY* genes in *E. fulvus*. Red lines are WRKY gene pairs genomes between plant genomes. Gray lines are collinear blocks for interplant genomes.

**Figure 6 genes-13-02102-f006:**
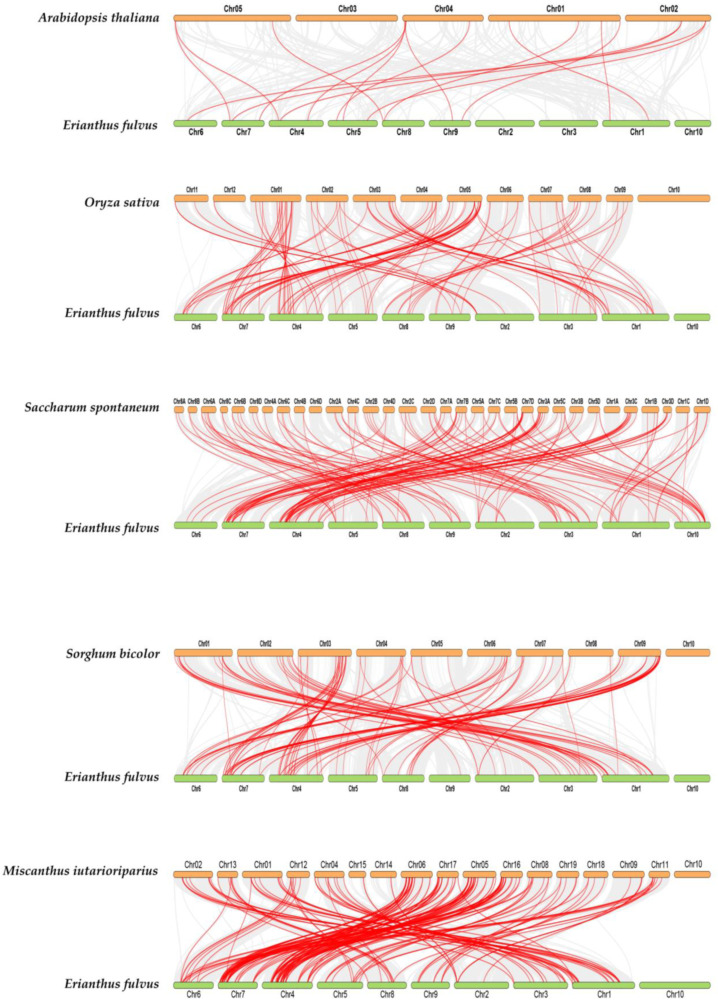
Synteny analysis of *WRKY* family. *A. thaliana*, *S. bicolor*, *M. iutarioriparius*, *E. fulvus*, *O. sativa*, and *S. spontaneum*. Red lines are the syntenic *WRKY* gene pairs. Gray lines are collinear blocks.

**Figure 7 genes-13-02102-f007:**
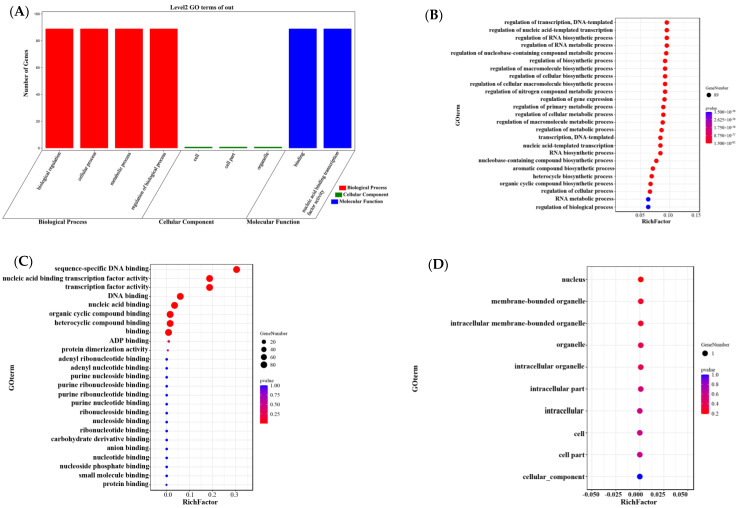
Gene Ontology (GO) annotation. (**A**) Direct count GO. Red bars represent biological processes (BP), blue bars represent molecular functions (MF) and green bars represent cellular compartments (CC); (**B**) Major Biological Process (BP); (**C**) Major Molecular Function (MF); (**D**) Cell Compartment (CC).

## Data Availability

Data are contained within the Appendix A.

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
