# Peer review of "Bioinformatics Analysis of WRKY Family Genes in Erianthus fulvus Ness"

_genes, 2022, doi:10.3390/genes13112102_

Round 1

Reviewer 1 Report

The manuscript entitled "Bioinformatics analysis based on the genome-wide WRKY gene family of Erianthus fulvus Ness" is a nice attempt to study "Genome-wide distribition and in-silico characterization of WRKY gene family of Erianthus fulvus", in a well-structured manner. Still there are some important questions unanswered like, the aim of the work is not clear, what are the objectives of the work etc. The title of the manuscript also needs improvement describing objectivity. The introduction section needs to be revised. Method, Materials, discussion and conclusion have been well written.

Line No. 84-89: In the introduction section the sentence started from “In addition” should be clarified. The paragraph should be revised.

Line No. 100-101: The (E-value 10-5) should be (E-value 10-5)

Have the plant transcription factor database (http://planttfdb.gao-lab.org/) was consulted for the study or not? Have the EfWRKY01-EfWRKY89 sequences were submitted in any database?

The validation of results was performed or Not?

Author Response

请参阅附件。

Reviewer 2 Report

WRKY transcription factors are important transcriptional regulators in plants. Genes in WRKY family are involved in many biological processes including secondary metabolism, phytohormone signaling, and defense response to biotic and abiotic stresses. In this manuscript, the authors identified the WRKY genes in E. fulvus Ness. 89 gene members were identified. The authors performed a series of bioinformatics analyses such as secondary structure prediction, phylogenetic tree construction, gene structure analysis, chromosome localization, gene duplication, regulatory elements analysis, and GO annotation. Overall, the data presented was potentially informative. However, the manuscript was in low quality and needs to be improved.

1.     In the method section, the authors listed most of the software names, however, the implementation details were lacking, and the parameter settings that were used should be described in this section to ensure reproducibility. For example, SignaIP-5.0 and SOPMA in section 2.2; MEME and TBtools in section 2.4;

2.     Line 177: “The proteins in Group II were further divided into five subgroups G; IIa-e” How were these subgroups divided? Was this based on conserved domains and/or motifs?

3.     Section 3.3: the authors capitalized and spaced the group names alternatively.  For instance, “group II” in line 175, “GroupII” in 186, “GroupI” in line 184, and “G;IIa-e” in 178. Please keep them consistent throughout the manuscript.

4.     Line 182-186: “with EfWRKY07 in Group I and Group III in The conserved structures of WRKY in EfWRKY31, EfWRKY45, EfWRKY50, EfWRKY54 and EfWRKY83 in Group became WRKYGEK; the conserved structures of WRKY in EfWRKY40, EfWRKY53, EfWRKY57 and EfWRKY67 in Group all became WRKYGKK”. This sentence was very hard to understand, please rephrase it.

5.     Line 186-187: “This phenomenon is consistent with the variation in rice.” How was this consistent with rice? Did the author analyze the rice data? Otherwise please add the reference.

6.     Line 192-195: “distribution of conserved motifs and gene structure analysis revealed that the motif1 motif was highly conserved, except for EfWRKY51, The distribution of conserved motifs and gene structure analysis revealed that motif 1 was highly conserved, except for three genes” This sentence did not make sense.

7.     Figure 2A, the resolution of this figure was too low, it was hard to see the details.

8.     Figure 2A, the motif annotation was not clear. Some motifs were in similar colors, for example, motifs 3 and 5; motifs 1 and 7.

9.     Figure 2, the gene length in 2A and 2B were not consistent. For example, EfWRKY07 was longer than EfWRKY06 in 2A but it was shorter than EfWRKY06 in 2B.

10.  Figure 2, “green boxes indicate CDS, and yellow boxes indicate UTR.” In the figure green was UTR and yellow was CDS.

11.  Line 213,” There are 11 WRKY genes of known function in the EfWRKY gene” Were these 11 known WRKY genes OsWRKY or EfWRKY?

12.  Line 217-219: “members of Group IIa of the EfWRKY gene family members of Group IIa of the EfWRKY gene family 218 clustered with OsWRKY76 and OsWRKY71,” What did this sentence mean?

13.  Figure 4 needs more interpretation.

14.  Please add a detailed figure legend for Figure 5.  

Round 2

Reviewer 1 Report

The revised manuscript is in acceptable form only few typographical mistakes are there.

Reviewer 2 Report

The authors addressed most of the issues. I have a few general suggestions:

1. The authors may need to double-check the terminologies: for example, in line 238, "was constructed using the NJ (neighborhood method)", it should be "neighbor-joining" method.

2. The authors should follow the standard when referring to the gene and protein names in the manuscript. In general, symbols for genes are italicized (e.g.,OsWRKY89 gene), whereas symbols for proteins are not italicized (eg., WRKY proteins). In many places, the authors used italicized format alternatively.

3. To present the methods and results, the authors should keep using the past tense instead of the present tense. In many places, the authors used both which was confusing. For example "The E. fulvus genome files were swiftly compared with the gff3 files using the TBtools software(Parameter setting: CPU for BlastP is 2, E-value 10-5, Num of BlastHits is 10)."

4. The language should be polished before publication.
